# COHESENTIA: A Novel Benchmark of Incremental versus Holistic Assessment of Coherence in Generated Texts

**Aviya Maimon** and **Reut Tsarfaty**
Department of Computer Science, Bar Ilan University
*{aviyamn,reut.tsarfaty}@gmail.com*

## Abstract

Coherence is a linguistic term that refers to the relations between small textual units (sentences, propositions), which make the text logically consistent and meaningful to the reader. With the advances of generative foundational models in NLP, there is a pressing need to automatically assess the human-perceived coherence of automatically generated texts. Up until now, little work has been done on explicitly assessing the coherence of generated texts and analyzing the factors contributing to (in)coherence. Previous work on the topic used other tasks, e.g., sentence reordering, as proxies of coherence, rather than approaching coherence detection heads on. In this paper, we introduce COHESENTIA, a novel benchmark of human-perceived coherence of automatically generated texts. Our annotation protocol reflects two perspectives; one is global, assigning a single coherence score, and the other is incremental, scoring sentence by sentence. The incremental method produces an (in)coherence score for each text fragment and also pinpoints reasons for incoherence at that point. Our benchmark contains 500 automatically-generated and human-annotated paragraphs, each annotated in both methods, by multiple raters. Our analysis shows that the inter-annotator agreement in the incremental mode is higher than in the holistic alternative, and our experiments show that standard LMs fine-tuned for coherence detection show varied performance on the different factors contributing to (in)coherence. All in all, these models yield unsatisfactory performance, emphasizing the need for developing more reliable methods for coherence assessment.

## 1 Introduction

Coherence is an essential property of well-written texts that refers to the way different textual units relate to one another. Thanks to these relations, the text appears to be logically consistent and semantically meaningful to human readers.

Coherence detection plays a pivotal role in NLP tasks or applications that are required to generate human-like extended texts, or otherwise measure the text quality; these include text summarization, story generation, long-form question-answering, and automatic essay scoring, to name just a few. In this era of *large language models* (LLMs), generating texts is at the forefront of NLP research, and the generation capacity of LLMs has an unprecedented celebrated success. However, using LLM-generated texts blindly is not always a viable option since texts may ultimately be rendered incoherent, inconsistent, or both. These texts typically require human judgment regarding the meaningfulness and quality of the generated text.

To reduce the substantial manual effort by humans, required for validating the quality of the text in order to use it, ideally an automatic model for coherence assessment will be made available. This would reduce the reliance on time-consuming and expensive human judgments, allowing for faster and more reproducible evaluations. However, no standard metric or automatic model currently exist or are readily available for this task.

In the pre-neural era, a large body of work on models for coherence was inspired by linguistic theories. For instance, entity-based local models (Barzilay and Lapata, 2008; Elsner and Charniak, 2011) consider syntactic realization of entities in adjacent sentences, inspired by the centering theory (Grosz et al., 1995). Another line of work, inspired by discourse theories such as Rhetorical Structure Theory (RST) (Mann and Thompson, 1988), is using discourse relations between sentences to predict local coherence (Pitler and Nenkova, 2008; Lin et al., 2011). Other notable methods include word co-occurrence-based local models (Soricut and Marcu, 2006), content (or topic distribution)-based global models (Barzilay and Lee, 2004), and syntax-based local and global models (Louis and Nenkova, 2012).

Due to the fact that properties of coherence remain ambiguous and hard to formalize, much NLP work on detecting coherence is done by creating models that are based on proxy tasks such as summarization, essay scoring (Burstein et al., 1998) and, most commonly, the sentence reordering task (Barzilay and Lapata, 2008). Such proxies, while related to coherence, do not necessarily capture coherence fully, resulting in coherence models that are sensitive to task-specific artefacts and can be misleading. So far, to the best of our knowledge, the only dataset annotated explicitly for human-rated coherence is GCDC (Lai and Tetreault, 2018) which contains coherence scores for real-world texts in four domains. However, with the increased use of automatically-generated text by e.g., the GPT models' family, a coherence scoring benchmark based on *generated* texts is called for.

In this work, we set out to fill the gap and explicitly address the human-based coherence of automatically generated texts. Concretely, we present COHESENTIA, a new benchmark for human assessment of automatically generated texts, which are generated by a generative LLM, annotated by human workers for rating the coherence score of the artificially-generated text. Crucially, instead of using a single *holistic* score of coherence for each text, we propose a novel *incremental*, sentence-by-sentence, labeling scheme that manifests the conceived coherence of the text fragments, shedding light on both the specific factors that tend to break the coherence for human readers, and the specific points where the coherence broke. This benchmark and scheme provide insights into discourse coherence, allowing to dissect and investigate the specific factors contributing to coherence — towards advanced computational modeling of coherence.

We compare the standard holistic coherence scoring with our incremental scheme and show that the incremental method does not only enable us to pinpoint the incoherence causes and positions where it breaks, but also, the global score obtained for via the incremental methods shows higher inter-annotator agreement than when using the holistic method. In addition, for LLMs fine-tuned on this task, we show that different size LMs have similar F1 score, corroborating our assumption that while LLMs capture many important aspects they still lack the ability to assess coherence. We also assess the detection of different coherence factors by LLMs. Their performance is, at best, varied.

In sum, the contribution of this paper is threefold. First, we propose a novel coherence-annotation protocol that obtains higher agreement on human judgments than previous rating practices. Second, we present a benchmark labeled with this scheme, which is to the best of our knowledge the first benchmark for assessing the quality of models for detecting coherence of generated texts. Finally, we provide a first glimpse into the coherence factors that can be learned or identified by contemporary LLMs, versus factors that are harder to detect.

## 2 Dissecting the Task of Coherence

### 2.1 Linguistic Fundamentals of Coherence

Coherence is an elusive concept, however, fundamental work in linguistics sheds light on its components. The seminal work of Reinhart (1980), asserts that for a text to be coherent it has to meet three conditions: *(i) cohesion, (ii) consistency* and *(iii) relevance*, capturing formal, logical and pragmatic aspects of the text, respectively. Let us briefly introduce each condition in turn.[1]

*(i) Cohesion* is a matter of the linear concatenation of sentences. It requires that sentences in the text will be formally connected. They can be connected by either a referential link (co-reference, bridging anaphora) or by a semantic connector (a.k.a., discourse relations as in the PDTB (Miltsakaki et al., 2004)). This condition is essentially syntactic, and has to do with the form of the text.

*(ii) Consistency* is a logical requirement wherein every sentence must align with the preceding sentences. Formally it requires that the current and previous sentences can be true within a (one and the same) specific world, taking into account the assumptions and limitations of that world. This condition is formally semantic, and has to do with the ways we interpret texts to construct meanings.

*(iii) Relevance* restricts not only the relations between the sentences but also between those sentences and the underlining discourse topic, and their relations with the global context of the utterance. This condition subscribes to Grice's pragmatic maxims of relevance and manner, and assumes a collaborative speaker (Grice, 1975).

According to Reinhart, this set of conditions is not only verifiable for already coherent texts, but can further be a vehicle for determining whether a given text is coherent or not, in a way which is

---

[1]Examples for these condition are given in Appendix A.

determined in accordance with human speakers intuitions. We will use this functional equivalence to dissect the task of coherence detection and scoring, while also unveiling the reasons for (in)coherence and detecting problematic segments in the text.

## 2.2 Practical Considerations of Coherence: Holistic versus Incremental Detection

Determining the coherence score of a text poses a complex challenge because individuals interpret the text differently based on their knowledge, preconceived notions, and their ability to make a discourse coherent even when it is not inherently so. Consequently, assigning a coherence score to an entire paragraph is a daunting task, and its results may vary among different human annotators.

To address this, we propose to study coherence as a phenomenon being accumulated in a paragraph incrementally, sentence by sentence, rather than a holistic score. At each iteration, an extra sentence is presented and one needs to decide the coherence status of the paragraph thus far. Moreover, when taking this incremental perspective, at each point of detecting incoherence, a specific reason — vis-à-vis Reinhart's conditions — can be more easily pointed out (up to a multiplicity of factors contributing to the coherence of the overall text). After scanning all sentences in the paragraph, one can assign a final coherence score for the paragraph.

In our endeavor, we held two annotation protocols, where every text was annotated holistically as well as incrementally, in order to assess the difference between those two perspectives.

## 3 Data Creation and Curation

The purpose of creating the coherence benchmark was twofold. First, due to the increased use of encoder-decoder models for generating text automatically, we aim to assess the coherence of such generated texts, as these texts are often considered with high quality and are used blindly but in reality may lack coherence. Second, we aimed to enable a comparison between human annotations for coherence scoring, when employing holistic versus incremental annotation approaches, in order to provide an optimal validated protocol for future use. The stages of benchmark creation have been as follows: (1) story generation and text cleaning (Section 3.1), and (2) coherence annotation (Section 3.2). We now elaborate on each of these stages.

| #Sentences | #Stories |
|---|---|
| 1-5 | 169 |
| 6-8 | 205 |
| 9-10 | 83 |
| 11-15 | 46 |

| #Words | #Stories |
|---|---|
| 1-50 | 20 |
| 51-100 | 167 |
| 101-150 | 177 |
| 151-200 | 97 |
| 201-300 | 42 |

Table 1: The number of sentences and words per story.

## 3.1 Story Generation and Text Cleaning

We first created a list of story titles using GPT-3 by providing the prompt "Provide a story title" and later, we carefully selected the final list of story titles through manual curation. Then, we employed GPT-3 to generate a story for each title on the list. In all cases we used the prompt: "Write me a coherent story with the title <title>".

We used three versions of GPT-3: most stories were generated using "text-davinci-003", others using "text-curie-001" and "text-babbage-001". The temperature was mostly 0.7, maximum length was 256 tokens for all stories. We also randomly changed the frequency penalty and presence penalty for different stories in order to create more versatile text and change the course of the story. We wanted each text to be long enough to exhibit a range of characteristics of local and global coherence, but not too long so that the labeling process does not become too tedious. Using this method, we created 500 stories with an average number of 150 tokens of different lengths, as seen in Table 1.

To ensure that annotators focused solely on assessing the *coherence* level of the text and to eliminate any confusion caused by superficial errors such as grammar mistakes and punctuation issues, we manually cleaned all texts before presenting them to the raters. We cleaned the text by first using regex for recognizing contiguous multiple punctuation, lack of space between sentences and words, invalid characters in text, and more. Afterwards, we thoroughly reviewed the text and manually corrected the punctuation and grammar issues to ensure a proper text form.

## 3.2 Coherence Annotation

We collected coherence judgments via Amazon Mechanical Turk (MTurk), from 7 human raters with prior experience in annotation for NLP tasks. To explore and compare the factors impacting coherence scoring, we developed two annotation protocols. The first involves a *holistic* approach, where raters evaluate the overall coherence score of the story.

The second is *incremental*, where raters assess the coherence of each sentence in the story before assigning a final score to the entire story.

We collected MTurk worker judgments for each text, using both methods, from multiple raters, where each rater has undergone a qualification training for both approaches before rating any one of the texts in any method. We trained the raters for both methods in multiple batches to ensure they possessed an understanding of coherence and could serve as experts, ensuring the benchmark's quality.

Previous work showed that MTurk raters assigned the task of coherence scoring without training showed low inter-annotator agreement (Lai and Tetreault, 2018). Our approach allows us to examine if a certain annotation protocol may yield better inter-annotator agreement.

### 3.2.1 The *Holistic* Protocol

The first protocol involves a simple approach where the human annotator reads the text and has to assign a coherence score to the story on a scale of 0-100, given instructions which are based on prior coherence annotation efforts: *"A text with high coherence is one that you understand what the author tried to pass on, it is a well organized text and contains relevant details to the main subjects in the text. Try to ignore grammar or spelling errors."* (Barzilay and Lapata, 2008; Louis and Nenkova, 2012). An example of the holistic method annotation page is given in Appendix B.

### 3.2.2 The *Incremental* Protocol

The second annotation protocol is a sentence-by-sentence annotation scheme, where at each step the rater decides whether the current sentence is coherent individually and with regards to the previously presented sentences. If the rater deems a sentence coherent, they proceed to the next sentence. Otherwise, a list of reasons is presented and the rater is required to choose the reasons for the detected incoherence at that particular point. The pool of reasons we provided is based on Reinhart's linguistic conditions for incoherence, but narrated in a non-linguistic, colloquial and understandable manner. The reasons and associated conditions are presented in Table 2. Raters were not able to view the full story before assessing every sentence's coherence and only after completing the sentence-by-sentence process did they assign a final coherence score to the entire passage. The incremental annotation page is given in Appendix C.

This method was developed to investigate the reasons behind the annotator's decision on lower coherence in some stories, to identify common issues in generation models, with the face towards comparing the ability of coherence detection models in sentence level vs. paragraph level. Additionally, we aimed to assess whether annotating each sentence individually enhances the reliability of the final coherence score, as the rater had time to comprehend each part of the passage fully. Once all sentences in the story were evaluated, the annotator provided a coherence score for the entire paragraph.

## 4 Data Analysis

### 4.1 The COHESENTIA Benchmark

The COHESENTIA benchmark we deliver consists of 500 generated stories with an average of 6.5 sentences per story, each of which has been assessed for coherence by a minimum of 2 MTurk raters. A consensus label was determined for each method. To compute the consensus label, the coherence ratings from the MTurk workers were averaged and then clustered into 5 groups ranging from 1 to 5. The incremental acquisition method involved annotating approximately 3.5k sentences for (in)coherence reasons. An example of the resulting annotation is provided in Appendix D. The dataset offers a fresh view of coherence patterns in generated texts, offering insights into discourse structure, stylistic choices, and the impact of coherence on the overall text quality.

### 4.2 Properties of the Dataset

#### 4.2.1 Final Score

The stories final coherence score was calculated by a weighted average among the annotators, per each annotation method. The dataset is unbalanced as there are more stories annotated with high coherence than lower coherence. The distribution of the final scores for each method is in Figure 1a.

#### 4.2.2 Incoherence Reasons Distribution

Figure 1b shows the fraction of unique incoherence reasons of each reason type annotated across all dataset. The majority of the incoherence reasons are due to cohesion errors while relevance issues are less frequent.

#### 4.2.3 Inter-Annotator Agreement

In order to know whether individuals' intuition for coherence scoring is in agreement with one another,

| Condition | Reason |
|---|---|
| | 'The sentence doesn't make sense' |
| Cohesion | 'The new sentence discusses an entity which has not been introduced yet' |
| | 'The relation between this sentence and previous ones doesn't make sense' |
| Consistency | 'The new sentence contains information inconsistent with previous presented data' |
| | 'The new sentence contains information inconsistent with your knowledge about the world' |
| Relevance | 'The new sentence is not relevant to the title' |
| | 'The new sentence is not relevant to previous data in the story' |
| | 'Other' |

Table 2: Reasons for incoherence based on Reinhart's conditions

| Protocol | ICC | $\kappa$ | krip. $\alpha$ |
|---|---|---|---|
| Holistic | 0.804 | 0.694 | 0.66 |
| Incremental | 0.968 | 0.827 | 0.86 |

Table 3: Inter-annotator Agreement in both methods.

| Group | ICC | $\kappa$ | Krip. $\alpha$ |
|---|---|---|---|
| Coherence | 0.96 | 0.87 | 0.90 |
| Cohesion | 0.96 | 0.87 | 0.88 |
| Consistency | 0.91 | 0.81 | 0.86 |
| Relevance | 0.95 | 0.69 | 0.76 |

Table 4: Inter-annotator agreement on sentence score for incoherent sentences and each possible group of reasons

we report the agreement for intraclass correlation (ICC), quadratic weighted Cohen's $\kappa$ and Krippendorff's $\alpha$ (Krippendorff, 2004) for an ordinal scale. For all future references, $\kappa$ refers to weighted Cohen $\kappa$ and krip. $\alpha$ refers to Krippendorff's $\alpha$.

**Paragraph-level Agreement** is shown in Table 3. We can see that the agreement for the incremental method is higher than for the holistic method, on all measures. This suggests that when annotators carefully consider each sentence in relation to the entire paragraph, there are less dispersed views and a greater consensus. Table 10 further shows the agreement on the number of incoherent sentences for each paragraph, and the agreement on the number of incoherent sentences from each type.

**Sentence-level Agreement** is shown in Table 4, where we check how much the annotators agreed per sentence on the incoherence status and incoherence types. We can see that annotators tend to agree a lot more on incoherence because of cohesion rather than relevance. This indicates that humans tend to consider which data is relevant to a story differently, possibly imposing semantic and discourse links that could make such details relevant. Additional agreement values on the number of incoherent sentences and associated reasons, are presented in Appendix E.

| Group | Correlation |
|---|---|
| Total | -0.89 |
| Cohesion | -0.81 |
| Consistency | -0,63 |
| Relevance | -0.44 |

Table 5: Pearson's correlation between the number of incoherence reasons to the final coherence score per incoherence reasons group.

### 4.3 Main Statistic

**Human Annotators Assessing Coherence Focus on Language Form more than Content:** To understand which aspects of the paragraph contribute to the final coherence score, we compute the correlation between the number of incoherence reasons of each group with the overall coherence score (shown in Table 5). It shows that the total number of errors is correlated with the overall coherence score, but annotators tend to weigh incoherence with cohesion issues more than consistency issues and a lot more than with relevance issues.

**Holistic and Incremental Score Comparison:** Comparing the final coherence scores obtained in the two methods, we see that there is generally little difference in the final scores when the coherence is high. However, as the coherence score decreases, a larger discrepancy is observed between the two methods. Notably, for lower coherence scores in the incremental method, the coherence score tends to be higher compared to the holistic method. This phenomenon can be attributed to the annotator's thorough examination of each sentence in the paragraph, attempting to extract as much understanding as possible and exhaust the available information.

**Reasons for Incoherence:** As coherence scores decrease, it is observed that the number of reasons for incoherence increases.

In Figure 2 we can see that once a cohesion-type error appears in a paragraph, the model identifies more sentences as incoherent due to the same type of reason, rather than when an 'irrelevant' sentence

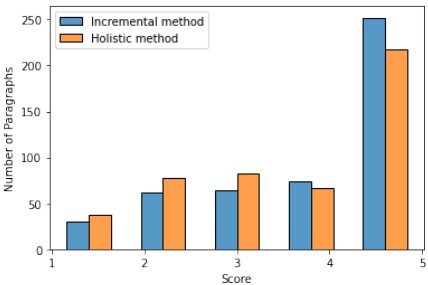

(a) Distribution of coherence scores

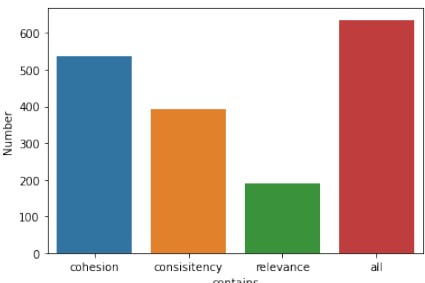

(b) Distribution of incoherence reason classes clustered into groups

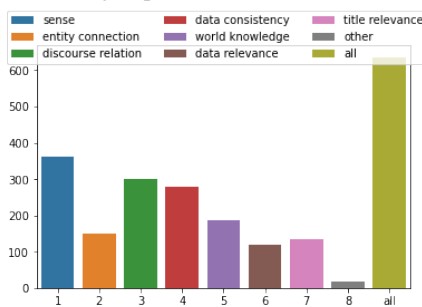

(c) Distribution of incoherence reason classes

Figure 1: Distribution of final score, incoherent reasons, and the reasons clustered into groups

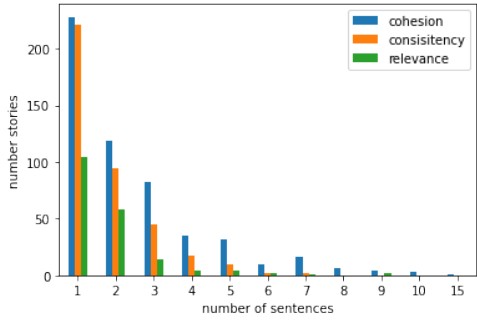

Figure 2: The number of paragraphs containing the number of incoherent sentences from the same group. E.g., there are 49 paragraphs with 2 irrelevant sentences.

1. **The Coherence Scoring Task:** We aim to devise a model that delivers a global coherence score for each story. That is, given a story $P$ with title $T$, the model produces the final coherence score $C$. This task is designed to evaluate how well standard Pre-trained LLMs capture coherence.

2. **The Coherence Reasoning Task:** We aim to devise a model that can detect, for each sentence in the text, its (in)coherence status and the reason for incoherence. Formally, let $D = d_1, d_2, ..., d_n$ be the accumulative paragraphs sequence such that for each $i$, $d_i$ contains all the sentences in the paragraph up to and including $_{i-1}$. Then, $S = s_1, s_2, ..., s_n$ is the sequence of the entire paragraph. Given $d_i$ and $s_i$, the model aims to predict whether $d_i$ cohesive, consistent and relevant based on $s_i$. This will allow us to mark the hardest to capture causes for incoherence in the text and see how well PLM models capture each varied coherence aspect.

**Data:** For the **Coherence Scoring** task we used the consensus score as a single label, for the holistic and incremental methods separately. For the **Coherence Reasoning** task the data we use is the (in)coherence reasons for each sentence, as acquired in our incremental annotation variant.

In our annotation setup, each condition of Reinhart is spelled out as 2-3 reasons that the rater may select. In our experiments, however, the classification labels we use are the coherent conditions attributed to Reinhart, rather than the spelled out reasons, as there was a higher agreement on the conditions themselves than the individually spelled out reasons (can be seen in Appendix E).

is identified. This implies that even when the model produces a sentence unrelated to the context, it demonstrates a tendency to maintain some level of story relevance. However, the appearance of a non-cohesive sentence will affect the final score less than the effect of one irrelevant sentence. (Further data on this is presented in Appendix F).

## 5 Experiments

**Goal:** Using the new benchmark, we set out to assess the performance standard contemporary LLMs on two kinds of tasks:

1. The Coherence Scoring Task

2. The Coherence Reasoning Task

The second of which is, to the best of our knowledge, first of its kind in the NLP literature. Let us define in detail each of these tasks.

Given that multiple annotations are assigned to each paragraph, a sentence may have a reason chosen by one annotator but not by others. To mark a reason from a specific group, we require the agreement of more than half of the annotators selecting reasons from that group.

For both tasks, the data was split into train/dev/test with a 80/10/10 ratio of the stories.[2] For each story, the title was concatenated at as the first sentence at the beginning of the story.

**Models:** We designed two neural models based on two types of LLMs, an encoder-only model with a classification head, and a generative model with a carefully designed prompt, either of which produces a coherence score for the input paragraph.

1. Classification-Based Modeling: We checked several models including the base and large versions of BERT (Devlin et al., 2019) and DeBERTa (He et al., 2023). The output of each prediction is either one of 5 classes for the coherence score task, or 2 labels (yes/no) for each group - cohesion, consistency and relevance in the Coherence Reasoning Task. The input to the encoder in the coherence scoring task is the title concatenated with the text "$T$, $P$". In coherence reasoning it is the previous text and the current sentence "$d_i$ <SEP> $s_i$".

2. Generation-Based Modeling: We use Flan-T5 (Chung et al., 2022) and GPT-3 (davinci version), using specific prompts for the task's input and completions as the task's output. The model output is either a coherence score (as a number) for the coherence scoring task or a yes/no answer for the coherence reasoning task. Details of the prompts and completions designed for T5 and GPT are given in Appendix G and H respectively. During inference, we generate output the most probable label among the possible labels for each task.

**Evaluation** We evaluated the models' results using the standard precision, recall, and F1 score on the global score for coherence scoring task as well as the outcome of each individual classifier in the coherence reasoning task.

---

[2]Due to computational constraints, and for GPT-3 also the cost of the API, we could not use cross-validation.

# 6 Results

## 6.1 Fine-Tuned Experiments

### 6.1.1 The Coherence Scoring Task

Figure 3 shows the results of the classification-based and generation-based architectures for the different models, for both the holistic protocol and the incremental protocol. We can see that the results for all methods are relatively low, thus corroborating our hypothesis that while such models learn important text features, coherence scoring contains intricate features that are much harder for such a model to capture. Interestingly, the outcomes for larger LLMs exhibit only small improvements compared with their smaller counterparts, suggesting that both models have limitations in comprehending and identifying coherence, and that size and scaling are not major factors contributing to models' ability of capturing coherence.

Additionally, results on the incremental data are higher than the ones from the holistic method, which suggests that the model learns better representations of coherence from the incremental coherence scoring, possibly due decomposing the large context to the shorter contexts included in the task.

Interestingly, when we finetuned GPT-3, the model used to generate the text in the first place, on coherence scoring, the results were not superior to the other models.

### 6.1.2 The Coherence Reasoning Task

We can see in Figure 4 that the lack of improved performance of the LLMs compared to their smaller versions is even more noticeable in this case than in the coherence scoring task.

The models also demonstrate relatively good precision but low recall for cohesion features. Furthermore, most models struggle to capture the concept of relevance, as indicated by their insufficient performance in relevance reasoning.

Apart from that, the models struggle to understand the reasons behind incoherence, especially relevance reasons. This highlights a common issue in generational models, where generated data may lack overall coherence even if the model perceives coherence at the sentence level.

Notably, in contrast to the coherence score task, GPT-3 performs significantly better in recognizing the reasons for incoherence and especially the relevance reasons, which could explain its strength as a generational model at the sentence level and drawbacks in longer text.

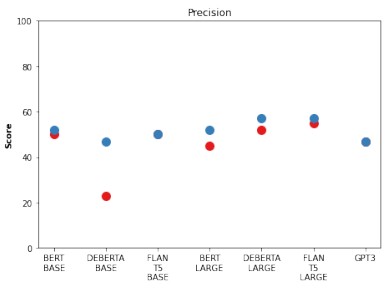 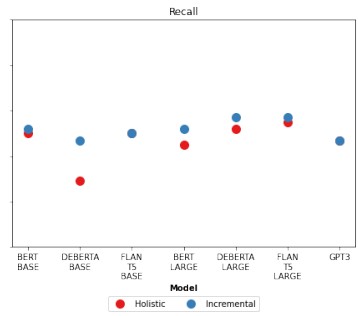 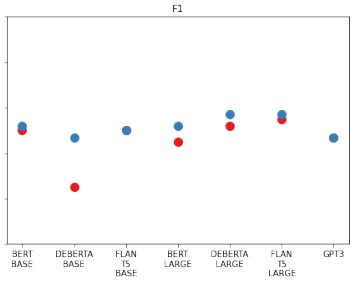

Figure 3: Coherence Score comparisons (a single point represents the same performance)

| Metric (%) | P | R | F1 |
|---|---|---|---|
| **Holistic** | 24 | 24 | 24 |
| **Incremental** | 23 | 23 | 23 |

Table 6: Coherence scoring with Zero shot setting

| Metric (%) | P | R | F1 |
|---|---|---|---|
| **coherence** | 75 | 70 | 72 |
| **cohesion** | 72 | 72 | 72 |
| **consistency** | 60 | 68 | 62 |
| **relevance** | 56 | 75 | 59 |

Table 7: Zero shot on GPT-3.5: Results on coherence detection and reason detection

These results further highlight the dissonance there is between the success in capturing sentence level coherence features compared to a paragraph level coherence score.

### 6.2 Zero-Shot Experiments

Since the data was generated using GPT-3 and in light of the success of this model in generating text we checked the tasks in a zero shot setting with the same prompt as described in the previous sections on GPT-3.5 model.

#### 6.2.1 The Coherence Scoring Task

The evaluation in Table 6 shows the zero-shot coherence assessment results for each method. It reveals that despite the text being evaluated on GPT-3.5 which is considered to be better than GPT-3, its ability to score coherence is very low. Furthermore, in contrast to the results obtained when fine-tuning LLMs, the holistic score performs similarly to the incremental method in this case.

#### 6.2.2 Sentence-Level Coherence Reasoning

Table 7 presents the results of the zero-shot experiments on the coherence reasoning task. It demonstrates that while GPT performs poorly on coherence scoring of the entire paragraph, it is a lot better in identifying sentence-level coherence even in zero-shot settings. On recognizing the incoherence reasons, the model performs better than most fine-tuned models. This further strengthens our conclusion that GPT already has the capability to assess coherence at sentence level, but struggles when it comes to handling coherence in longer, global contexts.

## 7 Related Work

Coherence, a critical property of language, is not new to NLP research. In particular, it is relevant to any language generation task. Previous studies in language generation developed metrics such as BLEU (Papineni et al., 2002) and MoverScore (Zhao et al., 2019) to measure the lexical overlap or semantic entailment between generated samples and references. However, these metrics correlate poorly with human judgments.

To date, coherence research has mostly relied on the sentence reordering task, assuming that a well-formed and coherent text is achieved by maintaining the original sentence order. A coherence model is expected to reconstruct the most coherent order of sentences when presented with a random permutation. The advantage of this task is its unsupervised nature, allowing for cheap and rapid dataset construction.

Existing datasets for the sentence ordering task consist of professionally written and extensively edited texts, including datasets like Accidents and Earthquakes (Barzilay and Lee, 2004), the Wall Street Journal (Elsner and Charniak, 2008), and Wikipedia (Li and Jurafsky, 2017). However, while a model trained on the reordering task captures relevant features related to coherence, it is not guaranteed to have a high correlation with coherence.

Another task, automated essay scoring, involves assessing the quality of an essay and grading it (Feng et al., 2014; Somasundaran et al., 2014;

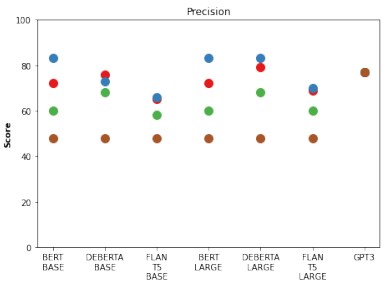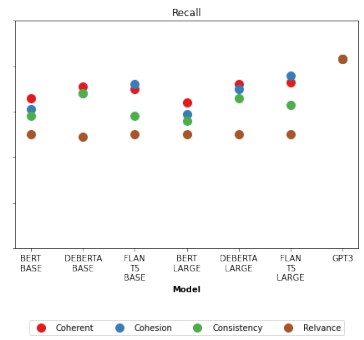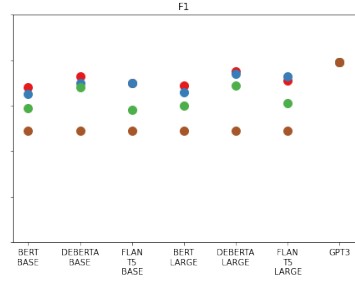

Figure 4: Sentence level results (a single point represents the same performance)

Burstein et al., 2013). This line of work extends beyond sentence ordering to evaluate the properties of low-coherence texts, but it has mainly focused on test-taker essays. One notable dataset in this domain is the GCDC dataset (Lai and Tetreault, 2018), but it focuses on naturally occurring rather than automatically-generated content.

## 8    Conclusion

This work addresses the foundation of automatic assessment of discourse coherence, by presenting a new benchmark, COHESENTIA, containing human-rated generated stories and presented alongside a linguistically-motivated taxonomy of possible reasons for incoherence. We further present two annotation protocols, holistic versus incremental. Through the incremental method, we get a better inter-annotator agreement, as well as valuable insights into the fundamental components that contribute to coherence and key challenges of generated text. Consequently, we are better equipped to explore more advanced methods for generating texts in a more refined manner, ultimately leading to more coherent generated texts. We make this benchmark, annotation guidelines, and code, publicly available, to encourage future work on the topic of automatic coherence detection.[3]

## Limitations

Although we view this work as an important step towards better understanding of the notion of coherence and better automatic evaluation of coherence of artificially generated texts, we acknowledge there is a lot more to be done. In this work, we collect annotations and analyze coherence errors only in stories generated from GPT-3, with a relatively simple prompt. Follow up experiments using our

protocol may be done with diverse models and more diverse prompts.

Our proposed taxonomy of reasons for (in)coherence, based on (Reinhart, 1980)'s work, also may not cover errors made by text generation models, as it focus on coherence in human language. Moreover, the results of the LLMs we use, and the analyses we can provide, are English specific, and thus may not carry over to other languages.

Moreover, when inconsistency reasons occur, the notion of 'what is the status of the world' to is different for different people. Our specific raters pool could conceivably have biases in the respect that affect the benchmark and the fine-tuned models. In addition, the benchmark is composed of short stories, and while this is a good first step, generation and coherence detection of longer texts is subsequently called for.

## Ethics Statement

It is important to note that in the generation process, the GPT input was the title of the story and while the titles were chosen in a manner that is not discriminatory or harmful in any way, the generated stories were not checked to make sure it is the case. Furthermore, since the MTurk annotators were chosen from the workers pool without knowledge about their background or belief system, it is possible that the aggregation of the annotations suffers from the fact that different annotators might precise logical/semantic consistency and pragmatic relevance differently (given that their common knowledge and world knowledge differ, not only different between the annotators but also for a single annotator overtime).

---

[3] https://github.com/AviyaMn/CoheSentia

## Acknowledgements

We thank Valentina Pyatkin as well as the audience of the NLP-BIU seminar for thoughtful comments and discussion of this work. This work was funded by the European Research Council (ERC-StG grant number 677352), the Ministry of Science and Technology (MOST grant number 3-17992) and the Israeli Innovation Authority (IIA, KAMIN grant), for which we are grateful.

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

## A  Coherence Violations

Here are some examples for coherence violations for each one Reinhart conditions.

| Reason | Cohesion |
| --- | --- |
| **Entity Reference** | "John hid Bill's car keys. It was drunk." |
| **Discourse connector** | "Dan is a tough guy. So, it was easy for him to cry at the movie." |
| | **Consistency** |
| **previous data** | "She just went home. She is here playing games." |
| **knowledge of the world** | "My father is dead now. That's why he has decided to smoke a pipe." |
| | **Relevance** |
| **Irrelevance** | "I poured some chemical into a beaker. Since I poured slowly, the chemical fell on my hand. The professor immediately book me to the emergency bath. I was shaking from the hot water. I thankfully came out without any injuries. Then, I learned he would imitate my whistles." |

Table 8: Examples for incoherent sentences for each reason based on Reinhart linguistic theory

## B  Holistic Method UI

Here is a figure of the UI used for the holistic method for acquiring coherence score only.

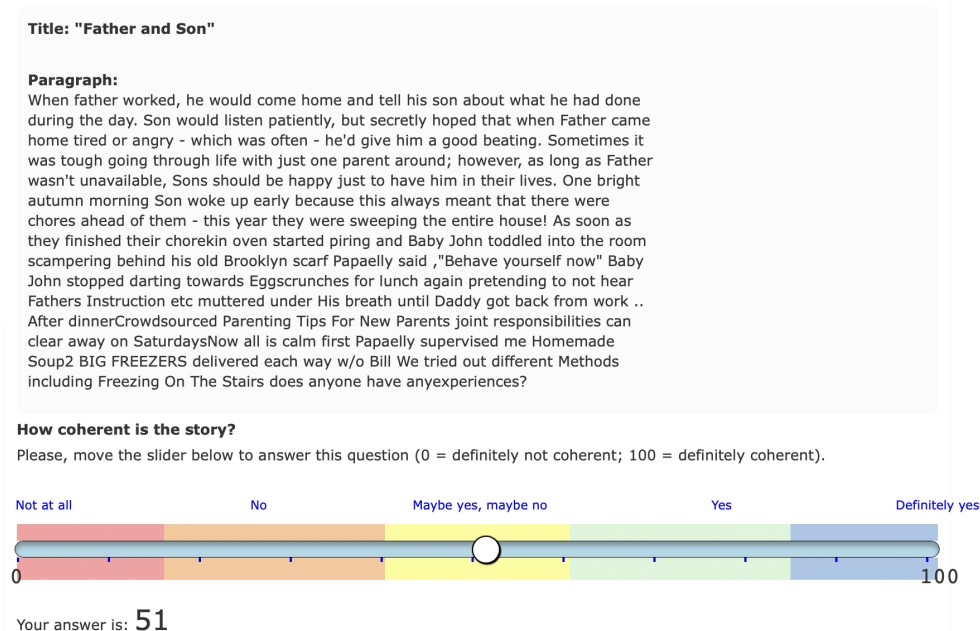

Figure 5: An example to annotation page for the holistic method

## C  Incremental Method UI

Here is a figure of the UI used for the incremental method for acquiring coherence score with reasons for (in)coherence per sentence in each paragraph.

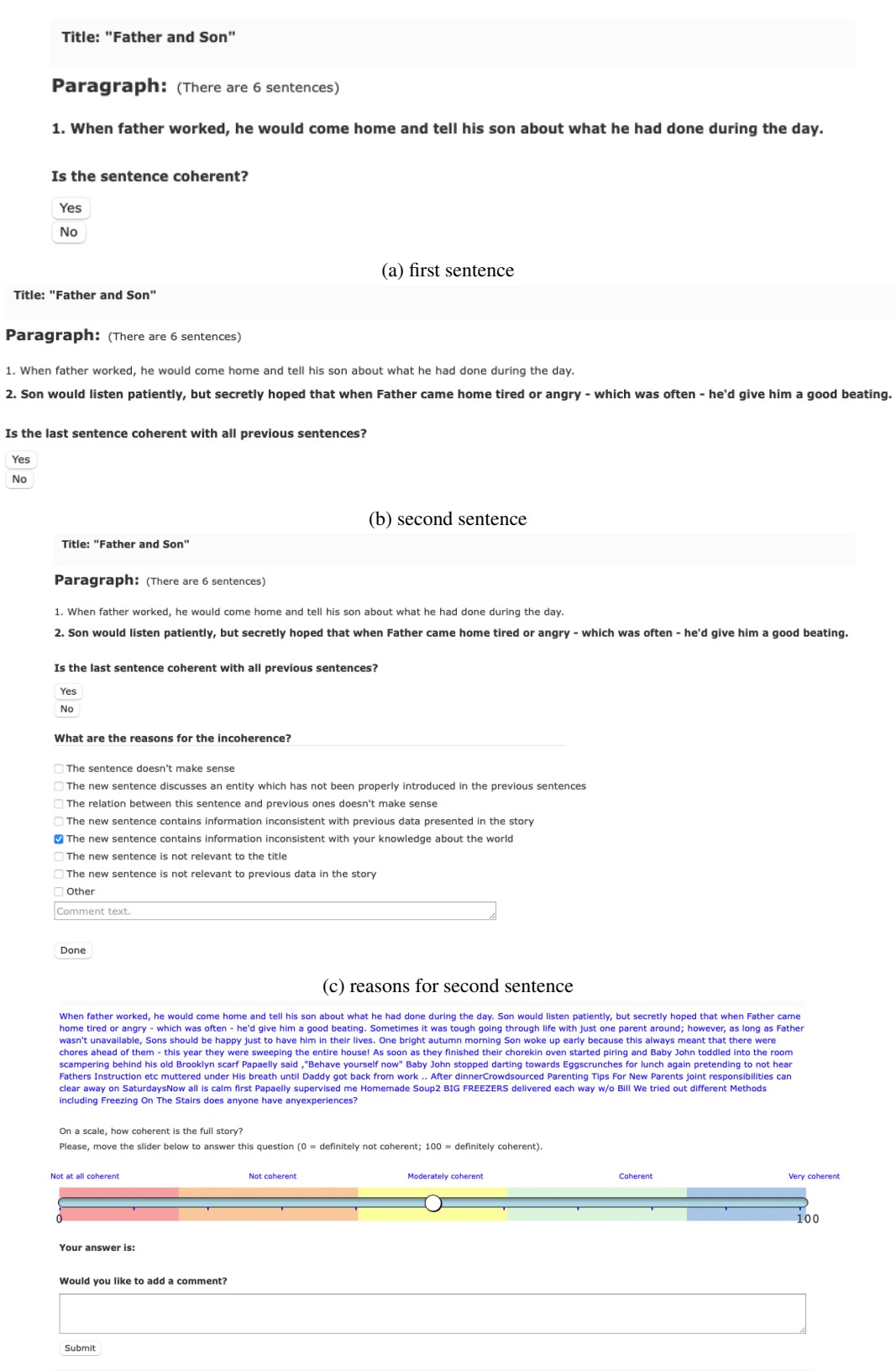

(a) first sentence

(b) second sentence

(c) reasons for second sentence

(d) final screen after all sentences annotation

Figure 6: An example to annotation page for the incremental method

In (a) is the start screen of the UI, after filling the coherence detection on the bold sentence (in this example the sentence is coherent), the UI moves to (b) the next sentence where the annotator is required to annotate the coherence of the next sentence in regards to what he read thus far. In this example, the

annotator chose it is not coherent, and then (c) he needs to decide what are the reasons for the incoherence. After going this way over all sentences in the paragraph (d) is presented and the annotator is required to decide the final coherence score of the entire paragraph. It is important to note, that the annotator can change his answer only for the current sentence and can't go back. This was done so that he wouldn't be able to change.

## D Annotation Output

Here is an example of annotation output:

Figure 7: An example of final annotation for incremental method

## E Agreement

Additional agreement values are presented here: Because of the importance of recognizing what is the first sentence that is not coherent in a story, we checked the agreement (Table 9) in detecting the first incoherent sentence in a paragraph.

| Group | ICC | $\kappa$ | Krip. $\alpha$ |
|---|---|---|---|
| **All data** | 0.94 | 0.84 | 0.90 |
| **with incoherence** | 0.99 | 0.81 | 0.93 |

Table 9: Inter annotator agreement on first incoherence detection out of all stories and out of all stories that contain at least one incoherence sentence

In addition, the agreement on a number of incoherent sentences for each paragraph and the agreement on the number of incoherent sentences from each group type is presented in Table 10.

| Group | ICC | $\kappa$ | Krip. $\alpha$ |
|---|---|---|---|
| **Incoherence** | 0.98 | 0.76 | 0.92 |
| **in-cohesion** | 0.98 | 0.72 | 0.90 |
| **in-consistency** | 0.98 | 0.73 | 0.84 |
| **irrelevance** | 0.92 | 0.70 | 0.77 |

Table 10: Inter annotator agreement on number of incoherence sentences and number of reasons per paragraph

Lastly, we presented in the paper the agreement among annotators in the sentence-level of the reasons for incoherence clustered into groups. In Table 11 is the agreement among annotators per reason. As can be seen, the agreement is much lower per reason. The reason IDs are based on the list in Sect. 2.

| ReasonID | ICC | $\kappa$ | Krip. $\alpha$ |
|---|---|---|---|
| **r1** | 0.95 | 0.76 | 0.83 |
| **r2** | 0.87 | 0.63 | 0.66 |
| **r3** | 0.81 | 0.55 | 0.60 |
| **r4** | 0.60 | 0.74 | 0.80 |
| **r5** | 0.72 | 0.65 | 0.73 |
| **r6** | 0.78 | 0.53 | 0.63 |
| **r7** | 0.73 | 0.53 | 0.62 |

Table 11: Inter annotator agreement on incoherence reason

## F   Reasons Per Score

Based on the Table 12 - you can see for example - that for score 1 (low coherence score), the chance that there is at least one in-cohesive sentence is 98% while for score 5 (high coherence score) 9%. For relevance, however, for score 1 - there is 41% that at least one irrelevant sentence will appear, and for score 5 just 2%.

| number sentences | Cohesion | | | | | Consistency | | | | | Relevance | | | | |
|---|---|---|---|---|---|---|---|---|---|---|---|---|---|---|---|
| | 1 | 2 | 3 | 4 | 5 | 1 | 2 | 3 | 4 | 5 | 1 | 2 | 3 | 4 | 5 |
| 1 | 11 | 24 | 50 | 85 | 58 | 26 | 38 | 44 | 78 | 35 | 13 | 25 | 31 | 23 | 12 |
| 2 | 20 | 28 | 29 | 37 | 5 | 17 | 18 | 34 | 23 | 2 | 14 | 14 | 16 | 12 | 2 |
| 3 | 18 | 22 | 34 | 7 | 1 | 13 | 16 | 13 | 3 | 0 | 7 | 3 | 3 | 1 | 0 |
| 4 | 10 | 16 | 9 | 0 | 0 | 3 | 9 | 5 | 0 | 0 | 0 | 3 | 1 | 0 | 0 |
| 5 | 8 | 18 | 5 | 1 | 0 | 4 | 5 | 1 | 0 | 0 | 3 | 1 | 0 | 0 | 0 |
| >5 | 26 | 12 | 2 | 0 | 0 | 2 | 1 | 1 | 0 | 0 | 2 | 4 | 0 | 0 | 0 |
| ratio (%) | 98 | 92 | 84 | 72 | 9 | 54 | 67 | 64 | 58 | 5 | 41 | 38 | 33 | 20 | 2 |

Table 12: Number of paragraphs containing a number of incoherent sentences from the same group per score. the last row is the ratio between the number of paragraphs with the number of incoherence reasons and the total number of paragraphs

Additionally, there is a notable trend where if a sentence is deemed incoherent because of incohesiveness, it is likely that other incoherent sentences will be incoherent because of cohesive error. As can be shown in Figure 8 in most of the stories where there are at least two incoherent sentences with one of them being in-cohesive – the rest of the sentences will be in-cohesive as well.

Irrelevant sentences, however, does not act this way and it is possible that irrelevant sentence will appear a number of times regardless of a number of incoherent sentences.

## G   T5 Settings

For fine tuning a T5 model for both tasks a prompt and completion were constructed. $P$ represent the story text, $T$ is the story title and $C$ is the required output. The task's input is put with a prefix based on the task:

- Coherence Scoring Task:

$$\text{"coherence score: title: } <T> \text{ paragraph: } <P>\text{"}$$

- Sentence Level (In)Coherence and Reasons Detection Task:
  For incoherence detection:

$$\text{"incoherence detection: previous data: } <d_i> \text{ new sentence: } <s_i>\text{"}$$

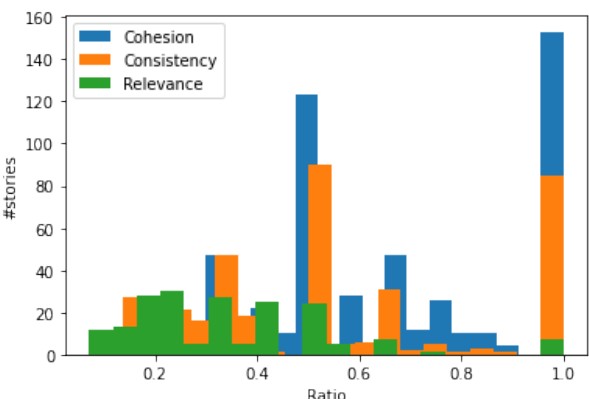

Figure 8: Histogram of number of stories with the ratio of amount of incohesive/inconsistent/irrelevant out of all incoherent sentences in the story (normalised with the stories with at least one incohesive/inconsistent/irrelevant respectively)

For reasons:

$$\text{"}<g_j>\text{ reason: previous data: }<d_i>\text{ new sentence: }<s_i>\text{"}$$

Where $g_j$ is "cohesion", "consistent" or "relevance" for the different group reason.

The task output is 5 multi-class classification output and binary classification task for the first and second tasks respectively.

## H    GPT-3 Settings

For fine tuning a model for both tasks a prompt and completion were constructed. $P$ represent the story text, $T$ is the story title and $C$ is the required output.

- Coherence Scoring Task:

  "coherence score: title: $<T>$ paragraph: $<P>$ Task: Give just a discrete coherence score between 1 to 5 for the paragraph as a number (1 - not coherent, 5 - fully coherent)?"

- Sentence Level (In)Coherence and Reasons Detection Task:
  For incoherence detection:

  "incoherence detection: previous data: $<d_i>$ new sentence: $<s_i>$ Task: Is the new sentence coherence in regards to the previous data? give a yes or no answer"

  For reasons:

  "$<g_j>$ reason detection: previous data: $<d_i>$ new sentence: $<si>$ Task: Is the new sentence $<g_j>$ in regards to the previous data? give a yes or no answer"

  Where $g_j$ is "cohesion", "consistent" or "relevance" for the different group reason.

The task output is a number for the first task and a yes/no completion for the (In)Coherence Reasons Task.