# OpenReview forum: "COHESENTIA: A Novel Benchmark of Incremental versus Holistic Assessment of Coherence in Generated Texts"
_EMNLP/2023/Conference — EMNLP 2023 Main_

### Official Review · Reviewer_8DsP · 2023-07-20

**Soundness:** 4

**Excitement:**

4: Strong: This paper deepens the understanding of some phenomenon or lowers the barriers to an existing research direction.

**Paper Topic And Main Contributions:**

The paper presents a novel benchmark for assessing coherence in automatically generated texts. The authors, together with the novel dataset, introduced a two-fold labelling schema: the holistic setting, where each text is featured with a global coherence score (i.e. considering the text as a whole), and the incremental setting, where each sentence of the paragraph is provided with a coherence score together with the reasons for incoherence of the fragment. The benchmark consists of 500 automatically generated texts annotated by humans through the Amazon Mechanical Turk system.
The annotators were requested to provide coherence scores for both experimental settings: in the holistic protocol the reader is asked to assess the coherence of the whole text by providing a score in the range 0-100 while in the incremental protocol the rater decides whether the current sentence is coherent individually and with regards to the previously presented sentences; if the annotator deems a sentence incoherent, a list of reasons is presented and the rater is required to choose the reasons for the detected incoherence at that particular point.
Finally, the authors assessed two different types of language models on the novel benchmark: encoder models provided with a classification head (BERT and DeBERTa), and generative models (Flan T5 and GPT3).
The results point out that results on the incremental setting are higher than those obtained on the more classical holistic setting. Additionally, the performance of large language models seem to be only a small improvement over the lower-sized models.

**Questions For The Authors:**

1. Will the dataset be released? I could not find any related statement.
2. In the Experiments section, is there a rationale for not using a tenfold cross-validation? I can imagine that this is due to the actual cost of the API and/or computational reasons. If so, it might be helpful for the reader to make the rationale behind the split decision explicit.
3. In both Figure 3 and Figure 4, GPT is associated with a single point. Is this because the performance is the same (holistic vs incremental in Figure 3 and coherent vs incoherent for Figure 4)?

**Reasons To Accept:**

The topic is interesting, the paper is well-written and easy to follow. The paper is self-contained. The authors provide clear motivations underlying the work. The paper introduces a novel benchmark that could be easily plugged in a framework designed to assess Large Language models so as to provide linguistically grounded intuitions on how such models  understand text coherence.

**Reasons To Reject:**

I feel no reason to reject the paper.

**Reproducibility:**

4: Could mostly reproduce the results, but there may be some variation because of sample variance or minor variations in their interpretation of the protocol or method.

**Reviewer Confidence:**

3: Pretty sure, but there's a chance I missed something. Although I have a good feel for this area in general, I did not carefully check the paper's details, e.g., the math, experimental design, or novelty.

**Typos Grammar Style And Presentation Improvements:**

Please find some minor comments below:
- please consider rewriting she on line 321 as she/he;
- please consider rewriting appendix on line 321 as Appendix;
- please consider rewriting cohen k on line 372 as Cohen k;
- In Figure 4, should the last model be GPT3?
- please consider rewriting si on lines 845, 847, 859 and 862 as s_i.

---

> ### Author Rebuttal · Authors · 2023-08-29
>
> We thank you for the positive feedback and we are glad you found the benchmark one “that could be easily plugged in a framework designed to assess LLM” as this is an important part of the role of these data we envision in the field.
>
> Regarding your questions:
> 1. Re: “Will the dataset be released?“
> Answer: Of course, the dataset along with all the necessary materials required for reproducibility will be made openly available once the paper is published. In the camera ready version, a link containing the dataset as well as the full code will be added to the paper. (We provide the anonymized version of these materials in the supplementary material of this submission).
>
>
> 2. Re: “In the Experiments section, is there a rationale for not using a ten-fold cross-validation? I can imagine that this is due to the actual cost of the API and/or computational reasons.”
> Answer: Yes, you are right. The reasons are the cost of the API and computational constraints. We will explicitly clarify the rationale in the camera ready version.
>
>
> 3. Re: “ In both Figure 3 and Figure 4, GPT is associated with a single point. Is this because the performance is the same?”
> Answer: Yes, a single point does indicate the same results. We will clarify it in the caption.
>
> We will of course make changes in the paper regarding your style and presentation improvements notes.
> Thank you!

---

### Official Review · Reviewer_8yo1 · 2023-08-04

**Soundness:** 4

**Excitement:**

3: Ambivalent: It has merits (e.g., it reports state-of-the-art results, the idea is nice), but there are key weaknesses (e.g., it describes incremental work), and it can significantly benefit from another round of revision. However, I won't object to accepting it if my co-reviewers champion it.

**Paper Topic And Main Contributions:**

This paper proposes an annotated resource, COHESENTIA, concerned with explicitly assessing the coherence of generated texts and with the analysis of incoherence and its underlying factors. Texts in the benchmark were annotated in two ways: a holistic assessment (an overall evaluation scored in 0-100), and an incremental assessment, where human raters considered text sentence by sentence, in incremental fashion. In this setting, also the reasons at the base of incoherence judgements were annotated (based on literature, namely from Reinhart's conditions), which may provide helpful information to learning systems. Comparison between the two annotation schemes shows that the IAA is higher in the incremental setting. The work proposes an experimental section, the Authors fine-tuned LMs to detect incoherence.

Obtained accuracy is not fully satisfying ("Coherence scoring contains intricate features that are much harder for a model to capture"); another outcome is that "Larger LMs exhibit only small improvements compared to their smaller counterparts, suggesting that both models have limitations in comprehending and identifying coherence".

Also considering the reported experimental results I have a question, and I'd solicit Authors to elaborate on this point. LM are basically probability distributions over text sequences: as such, LMs are unfit to deal with logical reasoning or matters such as coherence. In this sense, one is not surprised for the poor perfomance attained in the evaluation Section. One would thus be curious to know how do authors expect that whatever LM trained on their data this may generalize to previously unseen stories. How a device such as a LM is expected to help in higher-order tasks, such as deciding whether 'The relation between this sentence and previous ones doesn’t make sense'?
Putting at the heart of the system a closed resource such as GPT* may be detrimental to investigate inner mechanisms and implementation details. Are the Authors willing to elaborate on these issues?


**Questions For The Authors:**

[Q1] LM are basically probability distributions over text sequences: as such, LMs are unfit to deal with logical reasoning or matters such as coherence. In this sense, one is not surprised for the poor perfomance attained in the evaluation Section. One would thus be curious to know how do authors expect that whatever LM trained on their data this may generalize to previously unseen stories. How a device such as a LM is expected to help in higher-order tasks, such as deciding whether 'The relation between this sentence and previous ones doesn’t make sense'?

[Q2] Putting at the heart of the system a closed resource such as GPT* may be detrimental to investigate inner mechanisms and implementation details. Are the Authors willing to elaborate on these issues?

**Reasons To Accept:**

- The paper proposes an interesting work on a relevant issue, that is explicitly assessing the coherence of generated texts. This is a relevant theme to NLP, since it lies at the base of our understanding of LMs and their generative features.


**Reasons To Reject:**

- The computational approaches employed in the experiments seem to require some further (higher-level) mechanism to deal with the task at hand: or, equivalently, the task might not be appropriate to be dealt with Language Models.

**Reproducibility:**

4: Could mostly reproduce the results, but there may be some variation because of sample variance or minor variations in their interpretation of the protocol or method.

**Reviewer Confidence:**

4: Quite sure. I tried to check the important points carefully. It's unlikely, though conceivable, that I missed something that should affect my ratings.

**Typos Grammar Style And Presentation Improvements:**

Minor issues & Typos

- References should be corrected: please check a guide on text references, such as https://cljournal.org/style_guide_refs.html
- Figure 1 is too tiny, nearly unreadable, should anybody ever try to print the paper...
- line 424 "In Figure 2 **you** can see that once a cohesion error appears" -> in scientific literature it is rather unusual that Authors directly refer to their readership; impersonal forms should be preferred -> "it can be seen" or even 'we can see' would be acceptable
- line 496: 'concatinated' -> 'concatenated'
- line 657: 'generatio' -> 'generation'
- line 825 broken reference: Sect. ??
- Figure 4 is too tiny, nearly unreadable

---

> ### Author Rebuttal · Authors · 2023-08-29
>
> Thank you for the time and effort, we appreciate your thorough evaluation of our paper.
> We are pleased to learn that you find our paper interesting and fundamental for understanding LMs’ generative aspects.
>
> Regarding your questions:
> 1. Re: “One would thus be curious to know how do authors expect that whatever LM trained on their data this may generalize to previously unseen stories. How a device such as a LM is expected to help in higher-order tasks, such as deciding whether 'The relation between this sentence and previous ones doesn’t make sense'?”
> Answer:
> You are right, LMs, being probabilistic, don’t focus the training on logical reasoning. That said, LLMs are frequently used for tasks such as generation and summarization, which entails production of longer texts that are at least in principle required to be inherently coherent. It has also been shown that LLMs, trained extensively, exhibit some logical reasoning. For example, the paper by Honovich et al 2021 (https://arxiv.org/pdf/2104.08202.pdf) shows how fine-tuning LLMs using NLI tasks improves their ability in capturing consistency.
> Hence, our benchmark facilitates the evaluation of generation-based models, which is vital for high-quality long generated texts. This paves the way for potential applications such as identifying incoherent sections for enhanced text generation or serving as a tool for human-in-the-loop editing.
>
>
> 2. Re: “Putting at the heart of the system a closed resource such as GPT* may be detrimental to investigate inner mechanisms and implementation details. Are the Authors willing to elaborate on these issues?”
> Answer:
> The fact that GPT* is a closed resource is part of the challenge that this dataset is in fact trying to mitigate. Specifically, while we can't access its internal details, the dataset aims to provide a methodological way to probe its resulting generated text for human-like discourse coherence. Crucially, we make this dataset and accompanying annotations available for coherence assessment of text generated by *any* model, and not necessarily by GPT*.
>
> We wish to kindly ask, in what sense you found the work incremental? Our purpose was to create a theoretical foundation (Reinhart) and a first-of-a-kind resource that allows to benchmark models for automatic coherence detection. The models we show are proof of the usability of this resource - this is a starting point for the community to improve upon, on this important topic.
> If there are particular elements that you find insufficient, we'd very much like to improve on that.
>
> We thank and acknowledge your comments on typos, grammar and other notes, which we will make sure to address in the camera ready version.
> We hope we addressed all of your concerns to your satisfaction.
>
> Thank you!

---

### Official Review · Reviewer_1hwA · 2023-08-07

**Soundness:** 5

**Excitement:**

4: Strong: This paper deepens the understanding of some phenomenon or lowers the barriers to an existing research direction.

**Paper Topic And Main Contributions:**

The paper "COHESENTIA: A Novel Benchmark of Incremental versus Holistic Assessment of Coherence in Generated Texts" introduces a dataset meant as a benchmark to asses coherence in generated texts. It discusses the linguistic aspects of coherence, where authors show a deep theoretical background, and presents two alternative and parallel annotation methods, holistic and incremental, showing that the latter produces higher agreement among annotators.


The paper is complemented with a set of experiments, organised into two subtasks (i.e. Coherence Scoring and Coehrence Reasoning), which show that the COHESENTIA benchmark is challenging enough to foster future work in enabling LLMs to handle and interpret coherence at a more convincing level.




**Reasons To Accept:**

The paper is well written, the methodology strong, and the gathered annotation provides a valuable tool for measuring the ability of automatic systems to model coherence.



**Reasons To Reject:**

No reasons to reject

**Reproducibility:**

5: Could easily reproduce the results.

**Reviewer Confidence:**

5: Positive that my evaluation is correct. I read the paper very carefully and I am very familiar with related work.

**Typos Grammar Style And Presentation Improvements:**


I have just some comments:
- move all refenrences into brackets
- neutral pronoun at 310 instead of "she"
- caption for figure 8 needs rephrasing
- few typos: ("otherwose"" at 045, "this" at 554, "(i)" at 496)

---

> ### Author Rebuttal · Authors · 2023-08-29
>
> Thank you for your supportive review, as well as the clear and accurate summary that implies that our submission was well-understood.
> The fact that you found our benchmark “challenging enough to foster future work in enabling LLMs to handle and interpret coherence at a more convincing Level” is exactly what we aimed for in constructing this dataset.
>
> We acknowledge your  suggested improvements related to typos and presentation and we will make sure to address them in the camera ready version.
>
> Thanks again!

---

### Meta-Review · Area_Chair_7oEX · 2023-09-08

**Recommendation:** 5

**Metareview:**

The paper presents a benchmark to assess human-perceived coherence (through crowd annotators on Amazon Mechanical Turk) in generated texts, which contains 500 automatically generated texts. The authors  discuss linguistic aspects of coherence including its deep theoretical background. They present two alternative annotation methods – holistic – the annotators were asked to provide a global coherence score for the whole text, and incremental – the annotators scored each sentence also providing a reason for the detected incoherence cases. The results show that the incremental method guarantees higher agreement among annotators.
Apart from the annotation procedures, the authors complemented the paper with further experiments organised into two subtasks  - Coherence scoring and coherence reasoning evaluating different language models using the resulting benchmark - encoder models (BERT and DeBERTa) and generative models (Flan T5 and GPT3). Here, the results were also better for the incremental setting. The results also show that performance of large language models seem to be only a small improvement over the lower-sized models.
Strong sides (pros): (1) definitely, an interesting topic for the community; (2) novel benchmark for coherence assessment which can be used in further studies, and also for texts generated with any model; (3) the presented methodology is sound; (4) the paper is clearly and well-written, it is self-contained.
I would not say that there are cons, but rather some flaws in the paper. They were all pointed in reviewers’ comments and include typos, formatting, etc.

---

### Decision · Program_Chairs · 2023-10-07

**Decision:**

Accept-Main

**Comment:**

The paper presents a benchmark to assess human-perceived coherence (through crowd annotators on Amazon Mechanical Turk) in generated texts, which contains 500 automatically generated texts. The authors  discuss linguistic aspects of coherence including its deep theoretical background. They present two alternative annotation methods – holistic – the annotators were asked to provide a global coherence score for the whole text, and incremental – the annotators scored each sentence also providing a reason for the detected incoherence cases. The results show that the incremental method guarantees higher agreement among annotators.
Apart from the annotation procedures, the authors complemented the paper with further experiments organised into two subtasks  - Coherence scoring and coherence reasoning evaluating different language models using the resulting benchmark - encoder models (BERT and DeBERTa) and generative models (Flan T5 and GPT3). Here, the results were also better for the incremental setting. The results also show that performance of large language models seem to be only a small improvement over the lower-sized models.
Strong sides (pros): (1) definitely, an interesting topic for the community; (2) novel benchmark for coherence assessment which can be used in further studies, and also for texts generated with any model; (3) the presented methodology is sound; (4) the paper is clearly and well-written, it is self-contained.
I would not say that there are cons, but rather some flaws in the paper. They were all pointed in reviewers’ comments and include typos, formatting, etc.